# Long-Term Effects of a Single Application of Botulinum Toxin Type A in Temporomandibular Myofascial Pain Patients: A Controlled Clinical Trial

**DOI:** 10.3390/toxins14110741

**Published:** 2022-10-29

**Authors:** Giancarlo De la Torre Canales, Mariana Barbosa Câmara-Souza, Rodrigo Lorenzi Poluha, Olívia Maria Costa de Figueredo, Bryanne Brissian de Souza Nobre, Malin Ernberg, Paulo César Rodrigues Conti, Célia Marisa Rizzatti-Barbosa

**Affiliations:** 1Clinical Research Unit (CRU), Centro de Investigação Interdisciplinar Egas Moniz (CiiEM), Egas Moniz, Cooperativa de Ensino Superior, CRL, 2829-511 Caparica, Portugal; 2Ingá University Center, Uningá, Paraná 87015-510, Brazil; 3Department of Prosthodontics and Periodontology, Piracicaba Dental School, University of Campinas, Piracicaba, São Paulo 13414-903, Brazil; 4Department of Dentistry, State University of Maringa, Paraná 87020-900, Brazil; 5Department of Dental Medicine, Karolinska Institutet, and the Scandinavian Center for Orofacial Neurosciences (SCON), 141 52 Huddinge, Sweden; 6Bauru Orofacial Pain Group, Department of Prosthodontics, Bauru School of Dentistry, University of São Paulo, São Paulo 17012-900, Brazil

**Keywords:** botulinum toxin type A, myofascial pain, temporomandibular disorders, muscle thickness

## Abstract

This study assessed the long-term effects of botulinum toxin type A (BoNT-A) in subjective pain, pain sensibility, and muscle thickness in persistent myofascial temporomandibular-disorder pain (MFP-TMD) patients. Fourteen female subjects with persistent MFP received BoNT-A treatment with different doses (10U-25U for temporalis muscle and 30U-75U for masseter muscle). The treatment was injected bilaterally in the masseter and anterior temporalis muscles in a single session. Clinical measurements included: self-perceived pain (VAS), pain sensibility (PPT), and muscles thickness (ultrasonography). Follow-up occurred 1, 3, 6, and 72 months after treatment for VAS and PPT and 1, 3, and 72 months for ultrasonography. For statistical analysis, the Friedman test with the Bonferroni test for multiple comparisons as a post hoc test was used for non-parametric repeated measures comparisons among the evaluation times. A 5% probability level was considered significant in all tests. VAS values presented a significant decrease throughout the study (*p* < 0.05). Regarding PPT values, a significant increase was found when comparing baseline data with post-treatment follow-ups (p < 0.05), and even though a significant decrease was found in muscle thickness when baseline values were compared with the 1- and 3-months assessments, no differences were found when compared with the 72 months follow-up (*p* > 0.05). A single injection of BoNT-A presents long-term effects in reducing pain in persistent MFP-TMD patients, and a reversibility of adverse effects on masticatory-muscle thickness.

## 1. Introduction

Myofascial pain (MFP) has been related to affect up to 95% of patients with chronic pain [1] and is reported by 45% of those with temporomandibular disorders (TMD) [2]. This condition is characterized by local and referred pain, muscle stiffness, and sensory changes, with a multifactorial etiology.

Considering its high prevalence and chronicity, multidisciplinary approaches have been suggested for treatment. Although non-invasive conservative treatments should be considered as the first-choice intervention [3], dealing with chronic pain conditions demands investigating new modalities to reduce symptomatology and provide quality of life. Therefore, botulinum toxin type A (BoNT-A) was introduced to be used in a variety of chronic pain conditions due to its antinociceptive effect, being reported to suppress neurotransmitters secretion, thereby decreasing peripheral and central sensitization [4,5].

Several studies have reported the positive effects of BoNT-A in pain syndromes [6], such as neuropathic pain [7], chronic migraine [8,9], trigeminal neuralgia [10], chronic back pain [11], chronic pelvic pain [12], and chronic myofascial pain in TMD patients [13]. Considering its use for TMD management, our/a previous study [13] has shown reduction in patient-reported pain and pain sensitivity, demonstrating BoNT-A’s positive effect for this condition. However, adverse effects were also reported, for example, worsened masticatory performance, decreased muscle thickness, and reduction in mandibular bone volume [13]. Such outcomes are mainly related to muscle paralysis, which led to changes in muscle and bone morphologies, that could, ultimately, impair daily activities.

However, studies evaluating improvements and adverse effects after BoNT-A’s applications in muscle pain conditions are limited to short-term follow-ups, which jeopardize findings regarding effectiveness [13]. In addition, no study has assessed if the reported adverse effects produced by BoNT-A are permanent or reversible. Therefore, the present study aimed to investigate the long-term effects of one single injection of BoNT-A on masticatory muscles, after 6 years, in patients with persistent MFP-TMD. The null hypothesis is that after all these years, no beneficial or negative effects should be found, with patients presenting pain scores and muscle thickness resembling the initial appointment.

## 2. Results

From the initial 60 patients treated with BoNT-A, 14 individuals (23%) returned for the 6-year assessment (Figure 1). Even though some investigators assessing the outcomes after 6 years of follow-up were different from the original study, interclass correlation (ICC) showed an excellent inter-rater reliability for all variables (0.912 for subjective pain intensity; 0.910 for pressure-pain threshold; and 0.915 for muscle thickness) [13].

### 2.1. Self-Perceived Pain: Visual Analog Scale (VAS)

Compared to the baseline data, a significant decrease of subjective pain was already found one month after treatment (*p* < 0.001). This pain reduction was maintained in subsequent examinations, with no significant differences until the 6-year evaluation (*p* > 0.05) (Table 1).

### 2.2. Pain Sensitivity: Pressure-Pain Threshold (PPT)

In the PPT evaluation for both muscles, there was no difference between the baseline data and the one-month evaluation (*p* > 0.05). However, after three months there was a significant increase in PPT’s values compared with the baseline and the one-month assessment period (*p* < 0.001) (Table 2). This significant difference remained until the last evaluation period, with no significant difference between the last three periods of assessment.

### 2.3. Ultrasound Imaging

Masseters and anterior temporalis muscles thickness during maximum volunteer contraction decreased after BoNT-A application, even after the first month (*p* < 0.001). Although this reduction remained after 3 months, at the 6-year period it was not possible to observe any statistically significant difference from the baseline (Table 3).

## 3. Discussion

To the best of the authors’ knowledge, this is the first clinical trial to demonstrate the improvement of pain features and the reversibility of masticatory muscles adverse effects, after a 6-year follow-up of a single injection of BoNT-A regardless of dosage, in persistent MFP TMD patients. The results showed that BoNT-A produced a significant reduction in subjective pain after the first month of evaluation, which remained until the 6-year evaluation. Likewise, BoNT-A increased PPT values of masticatory muscles since the third month of assessment, which lasted up to the 6-year follow-up. Conversely, even though BoNT-A decreased muscle thickness (an adverse effect) in the first two periods of evaluation, after 6 years there was masticatory-muscle thickness recovery.

The positive effects of BoNT-A in a variety of chronic pain conditions [5] have been reported by some well-designed clinical trials [7,8,10]. In addition, in vivo and in vitro studies have reported that the antinociceptive effect of BoNT-A is mainly related to a peripheral and central decrease of neurotransmitters such as glutamate, substance P, and calcitonin-gene-related peptide (CGRP) [14,15,16], and by its axonal transport to sensory regions of the trigeminal ganglion [17,18], the modulation of the spinal opioidergic or GABA-ergic system [19,20], and the prevention of microglia activation [21,22]. However, despite the fact that several clinical trials [23,24,25,26,27,28] have assessed BoNT-A efficacy for myofascial pain relief, its efficacy has not yet been established. Methodological shortcomings may explain the conflicting results. 

Our findings showed that a single injection of BoNT-A, regardless of the dose, already produced a significant improvement in terms of subjective pain after one month of evaluation and was sustained until the 6-year evaluation. Our findings are in line with reports demonstrating a decrease in subjective pain after BoNT-A injection in samples with the same characteristics as ours after 4 [26] and 6 [28] months of follow-up. Additionally, our study found that PPT values significantly increased after 3 months of assessment and that this effect was maintained until the 6-year follow-up. Likewise, previous studies [23,26,29] have shown that BoNT-A was able to diminish muscle tenderness to palpation when compared to the placebo over 3 to 6 months. On the other hand, a doble-blind crossover study [24] including the same population characteristics as ours showed that while BoNT-A achieved a clinically significant reduction in subjective pain (i.e., 30% less pain), no differences were found when compared with the placebo group after three months of evaluation of subjective pain intensity and PPT assessment. Perhaps the differences in the methodology, such as treating just one muscle group (masseter), may have influenced the results.

Moreover, one question remains unclear about the BoNT-A analgesic mechanism of action: is it predominantly peripheric or central? There is plenty of literature about the analgesic peripheral effects of BoNT-A [5]; however, the evidence and research about BoNT-A analgesic effects are increasing, leading to the proposal of new pathways that BoNT-A can use in order to reach the central nervous system. In animals, it was demonstrated that BoNT-A inhibits central nociceptor transduction and central neurotransmitter release [21,30]. In fact, a clinical trial involving neuropathic pain patients (chronic pain condition) showed pain improvement after BoNT-A injections, although the quantity of the neurotransmitter in the biopsies of painful regions remained high. Based on these results, the authors proposed that the analgesic effect of BoNT-A is mainly related to the central mechanisms of pain transmission, at least in chronic pain conditions. Since MFP is also considered a chronic condition, this hypothesis could also be applied to our positive results. Notwithstanding, it is also known that persistent MFP-TMD patients present a deficit in pain modulation [31]. Considering that BoNT-A modulates the opioidergic and GABA-ergic systems [19,20], its effects on pain modulation could also explain the long-term effects presented in our study, corroborating the later hypothesis.

BoNT-A treatment is considered generally safe because the doses used for pain conditions are far from the lethal doses. In addition, a systematic review [27] reported that this treatment is generally tolerated since self-reported minor adverse effects (e.g., short-term facial weakness, pain at the injection site, transient edema, and minimum discomfort during chewing) that resolved spontaneously were the most prevalent. However, the authors [27] also concluded that none of the included studies aimed to assess objectively adverse effects on muscle and bone tissue. Notwithstanding, in the last few years some clinical and experimental [27,32,33] studies have reported “severe” adverse effects of BoNT-A, which include masticatory muscle atrophy and the loss of mandibular bone volume. Our study confirmed literature findings since a significant reduction in masticatory muscle thickness was found after 1 and 3 months of injection, independent of dosage.

The reduction in muscle thickness is part of a series of events produced when BoNT-A is injected. First, a muscle paralysis is caused by a neural denervation that lasts for at least three months [34], during which new nerve fibers are produced to innervate the muscular portion that was paralyzed; however, it was demonstrated that these new motor fibers do not have enough potency to innervate the muscle effectively. Then, the lack of sufficient innervation causes structural changes in masticatory muscles, including the diminution of the quantity of muscle fibers [32], changes in the quality of remnant muscle fibers [32], the reduction in muscle-fiber size [33], the replacement of contractile tissue with fat [35], and an increase in the mRNA expression of muscle-atrophy markers [33]. Finally, as a result of all of the structural changes after BoNT-A injection, a decrease in muscle thickness is produced. Nevertheless, our study found that the decrease in muscle thickness seems to be reversable since significant muscle thickness recovery was found in masseter and anterior temporalis muscles after the 6-year follow-up, showing no differences with baseline data. This finding allows us to suggest that the recovery of masticatory-muscle thickness is not related to the units of BoNT-A injected, but it could be related to the number of times that BoNT-A is injected, as concluded by Rauso et al. (2022) [36]. A prospective interventional trial of patients receiving a protocol of three repeated cycles of 30U (considered in our study as low dosage) of BoNT-A for masseter hypertrophy for 6 months demonstrated a dramatic reduction in muscle thickness (42.52%) after a follow-up of 35 weeks, which was sustained after a 4-year follow-up [37]. This result supports the cumulative adverse effect of repeated cycles of low doses of BoNT-A on muscle thickness, which could be considered irreversible.

Although our results demonstrated positive effects of a single injection of BoNT-A regardless of the dose, caution is suggested when judging the present findings since some limitations should be mentioned. The present study was performed on a convenience population, without sex comparison. Additionally, although the rate of patients who returned to the 6-year follow-up could be considered acceptable (23%), a larger sample size is required for future studies. Additionally, it cannot be excluded that factors such as the placebo effect, the natural evolution of the disease (myofascial TMD pain), and the regression toward the mean may also partially explain treatment outcomes. However, it is important to mention that the study population was composed of persistent MFP TMD patients that had suffered for at least 6 months [13], and that BoNT-A was the only treatment that produced a significant improvement in their perceived pain. Moreover, it should be considered that patients could have received other treatments during the last 6 years; in fact, most of them reported that only self-care strategies to control pain (thermotheraphy and massage in the painful area) were used. Additionally, the risk of sample bias cannot be excluded, i.e., that the study population was mainly composed of patients that experienced significant positive effects after the BoNT-A injections performed in our previous study, a fact that certainly could have influenced the favorable effects reported in the present study. We recommend assessing the reversibility of adverse effects on bone tissues after a single injection of BoNT-A since it is related to changes in muscle tissues. Finally, future studies should assess the clinically meaningful adverse effects on muscle and bone tissue arising from multiple applications of BoNT-A, and especially their reversibility.

## 4. Conclusions

Based on the results and limitations of this study, and considering the possible placebo effects on pain variable, it can be concluded that:

A single injection of BoNT-A, regardless of dosage, could present long-term effects in reducing pain in persistent MFP TMD patients, that did not achieve significant pain relief using conservative therapies. 

There is a reversibility of adverse effects on masticatory-muscle thickness after a single injection of BoNT-A, which may be related to the number of times that the toxin is injected and not to the doses.

## 5. Methods

### 5.1. Subjects

The present clinical trial reports the 6-year follow-up results of a previous study [13] performed with 100 persistent MFP-TMD female subjects. The former study included BoNT-A injection, oral appliance (positive control), and saline solution (placebo) groups, and assessed outcomes before treatment and after 1, 3, and 6 months. Participants were recruited from women seeking TMD treatment at the TMD Clinic of Piracicaba Dental School, University of Campinas, São Paulo, Brazil. Patients were diagnosed according to the Portuguese version of the Research Diagnostic Criteria for Temporomandibular Disorders [38] by two calibrated researchers not involved in any other processes of the study (kappa coefficient = 0.80). The study was approved by the Research Ethics Committee of Piracicaba Dental School (CAAE # 22953113.8.0000.5418-Date: 4 February 2014) and the Brazilian Registry of Clinical Trials (ReBEC RBR-2d4vvv). All subjects were informed about the research purposes and provided written informed consent to participate in this clinical trial. Inclusion and exclusion criteria have been previously detailed [13].

Therefore, subjects included in the present study were a convenience sample of the study of De la Torre Canales et al. (2020) [13], which assessed the efficacy and adverse effects of different doses of BoNT-A up to six months of follow-up. Since the primary objective of this study was to assess the reversibility of adverse effects, the positive (splint) and negative (saline solution) control groups were not included, due to the absence of statistically significance differences in the previous one [13]. Therefore, only participants assigned to the BoNT-A groups (*n* = 60) were contacted. The patients were recruited via telephone for the 6-year assessment. 

### 5.2. Treatments

BoNT-A (100 U; Botox, Allergan, Irvine, California, CA, USA) was reconstituted using non-preserved sterile saline solution 0.9%. Bilateral intramuscular injections were performed using a 1 mL syringe with a 30-gauge and 13 mm needle. For each masseter muscle, doses from 30U to 75U were distributed in five sites 5 mm apart from each other and were injected in the inferior part of the muscle (on mandibular angle) after functional test (teeth clenching). For each anterior temporalis muscle, doses from 10U to 25U were distributed in five sites 5 mm apart from each other, which were determined according to the functional test considering the most prominent part, which had to be 1 cm external to the eyebrow. Injections were performed by a calibrated researcher during a single appointment. 

## 6. Primary Outcomes

### 6.1. Self-Perceived Pain: Visual Analog Scale

A visual analogue scale (VAS) [39] is a 100 mm horizontal line, anchored by the words “no pain” at the left end and “worst pain imaginable” at the right end. Participants were instructed to mark a line at any point representing the level of their current, worst, and average pain of the last month. Mean values were used for statistical analyses.

### 6.2. Pain sensitivity: Pressure-Pain Threshold 

Pressure-pain thresholds (PPT) [40] was measured using a digital algometer (model DDK- 20, Kratos^®^, Cotia, São Paulo, Brazil). The algometer has a 1-cm^2^ flat circular-shaped tip at one end, which was used to apply pressure to the most prominent part (determined with functional test) of the relaxed masseter and anterior temporalis muscles. The device was positioned perpendicularly to the muscles and applied with increasing and constant pressure of approximately 0.5 kgf/cm^2^ by a new single calibrated operator (Kappa = 0.89), which was blind for the injected groups. Patients were asked to press a button at the very beginning of pain sensation. The procedure was fully explained to each patient before the examination, and it was emphasized that the purpose of the study was to measure the PPT and not pain tolerance. PPT measurement sites were aligned with BoNT-A injections sites. The PPT was determined as the arithmetic mean of three measurements.

## 7. Secondary Outcome

### 7.1. Ultrasound Imaging 

The thickness of both masseters and anterior temporalis muscles during maximum voluntary contraction (MVC) was calculated using real-time ultrasound imaging (UI; SSA-780 A-APLIO Mx, 38 mm/7-18 MHz; Toshiba Medical SystemCo., Tokyo, Japan). A different calibrated operator performed the 6-year measurement. The patients were placed in a supine position at a height that ergonomically favored the examination. Muscle thickness was measured directly on the instrument’s screen, with an accuracy of 0.01 mm according to the most voluminous part of the muscle in the screen image. UI values were determined as the arithmetic mean of three measurements [41].

### 7.2. Data Evaluation and Statistical Analyses

Subjective pain intensity (VAS) and PPT were assessed at five-time points: before treatment (baseline) and at one, three, and six months, as well as six years after treatment. Data from muscle thickness, measured with ultrasound imaging, were assessed at four-time points: before treatment (baseline), one and three months after treatment, and six years after treatment. The inter-rater reliability was assessed by the Intraclass correlation coefficient (ICC). The Kolmogorov–Smirnov test demonstrated no normal distribution. Then, the Friedman test with the Bonferroni test for multiple comparisons as a posthoc test was used for non-parametric repeated measures comparisons among the evaluation times. All data were analyzed using SPSS Statistics 25.0 software (IBM^®^, New York, NY, USA). A 5% probability level was considered significant in all tests.

## Figures and Tables

**Figure 1 toxins-14-00741-f001:**
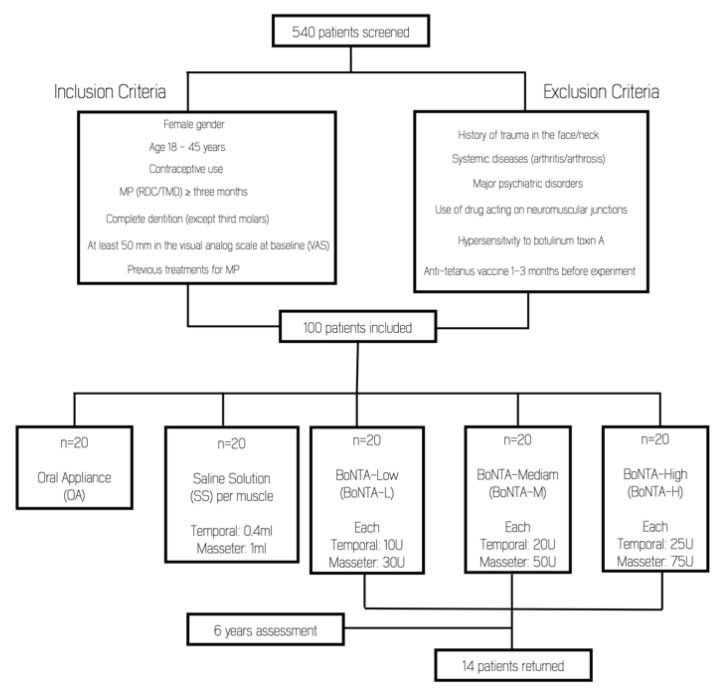
Flow diagram of patient’s recruitment.

**Table 1 toxins-14-00741-t001:** Median, minimum, and maximum values for subjective pain intensity (mm) in 14 patients with temporomandibular-disorder myofascial pain before and at different time-points after a single-injection of botulinum toxin.

	Time				
	Baseline	1 Month	3 Months	6 Months	6 Years
Median	7.85	1.15 *	0.00 *	0.00 *	0.30*
Minimum	5.90	0.00	0.00	0.00	0.00
Maximum	10.00	7.10	5.60	6.60	8.10

* Significant difference compared to baseline (*p* < 0.05; Friedman test with Bonferroni test for multiple comparisons as posthoc test).

**Table 2 toxins-14-00741-t002:** Median, minimum, and maximum values for pressure-pain threshold (Kg/f) in 14 patients with temporomandibular-disorder myofascial pain before and at different time-points after a single-injection of botulinum toxin.

		Time				
		Baseline	1 Month	3 Months	6 Months	6 Years
Anterior Temporalis	Median	0.51	0.89	1.02 *	1.22 *	1.18 *
	Minimum	0.30	0.26	0.32	0.45	0.56
	Maximum	0.75	1.44	1.62	1.62	2.05
Masseter	Median	0.45	0.85	0.95 *	1.07 *	1.03 *
	Minimum	0.26	0.29	0.32	0.44	0.53
	Maximum	0.79	1.42	1.54	1.28	2.03

* Significant difference compared to baseline (*p* < 0.05; Friedman test with Bonferroni test for multiple comparisons as posthoc test).

**Table 3 toxins-14-00741-t003:** Median, minimum, and maximum values for muscle thickness (mm) in 14 patients with temporomandibular-disorder myofascial pain before and at different time-points after a single-session treatment with botulinum toxin.

				Time	
		Baseline	1 Month	3 Months	6 Years
Anterior Temporalis	Median	2.02	1.30 *	1.38 *	2.15
	Minimum	0.80	0.80	0.90	1.30
	Maximum	1.70	3.20	2.40	3.40
Masseter	Median	12.52	10.42 *	11.37 *	13.20
	Minimum	9.95	7.60	8.95	12.00
	Maximum	12.35	12.70	12.45	15.60

* Significant difference compared to baseline (*p* < 0.05; Friedman test with Bonferroni test for multiple comparisons as posthoc test).

## Data Availability

The data presented in this study are available in this article.

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
