# Peer review of "Long-Term Effects of a Single Application of Botulinum Toxin Type A in Temporomandibular Myofascial Pain Patients: A Controlled Clinical Trial"

_toxins, 2022, doi:10.3390/toxins14110741_

Round 1

Reviewer 1 Report

Dear authors, i read your work with interest. Please find below my queries.

1) Few spell checks to correct on abstract. Please spell out the acronym MFP TMD.

2) Line 171-176. Please expand the possible effects on muscle thickness of multiple botulinum toxin injections and different doses and preparations.(DOI: 10.1016/j.jcms.2021.09.019)

3) Results tables layout is off. Please correct. Moreover, they are not clear presented expecially regarding the measurment unit used. For example: In table 1 are the values the mean millimeters? The same goes for the other tables. Please improve the redability and clarity.

4) As regards the ultrasound measurments: in line 260-261 you stated that the ultrasounds were performed by a "by a single calibrated operator". Do you confirm that even at the 6 years follow-up the operator was the same person? If this is the case, please stress this concept, since it will strenghten the validity of the results.

5) Since the original work divided the patients in 3 subgroups, have you considered to add a subgroup analisys (BoNT-A L/M/H)?

Best Regards

Author Response

Reviewer 1

Thank you for handling our manuscript. We revised it according to your comments and addressed each issue bellow, in a point-by-point manner. Our modifications are in track changes in the manuscript. The answers are in bold. We feel that by incorporating your suggestions in the revised version the clarity of our manuscript has improved. We would be glad to provide any further information needed.

1) Few spell checks to correct on abstract. Please spell out the acronym MFP TMD.

Reply: Thanks for the revisions. The spell errors were corrected in the abstract and in the entire manuscript, as well as the acronym was described.

2) Line 171-176. Please expand the possible effects on muscle thickness of multiple botulinum toxin injections and different doses and preparations. (DOI: 10.1016/j.jcms.2021.09.019)

Reply: Thank you for sending such interesting reference. The discussion was further improved based on this new information (lines 177-178).

3) Results tables layout is off. Please correct. Moreover, they are not clear presented especially regarding the measurement unit used. For example: In table 1 are the values the mean millimeters? The same goes for the other tables. Please improve the readability and clarity.

Reply: Thank you for the suggestion and carefully read of our manuscript. The tables were clarified.

4) As regards the ultrasound measurements: in line 260-261 you stated that the ultrasounds were performed by a "by a single calibrated operator". Do you confirm that even at the 6 years follow-up the operator was the same person? If this is the case, please stress this concept since it will strengthen the validity of the results.

Reply: We appreciate your observation. In fact, the 6-years outcomes were assessed by another researcher. This investigator was previously calibrated (kappa =0.89). Moreover, our research group uses a standardized protocol to guarantee the assessment of the same site even in different timepoints (Lines 261-262).

5) Since the original work divided the patients in 3 subgroups, have you considered to add a subgroup analysis (BoNT-A L/M/H)?

Reply: This is a nice question. Actually, we considered this subgroup analysis; however, the sample size included in the study did not allow to perform it. Moreover, since all L/M/H BoNT-A groups showed reduced muscle thickness (adverse effect) and lower pain report (positive outcome) in the previous evaluation, it would not be a bias having only one BoNT-A group.

Reviewer 2 Report

It is very timely and important to investigate the long-term effects of BoNT-A on patients, especially the long-term adverse effects. However, there are several flaws in the manuscript. First, the patient population is too small, and there is no placebo-controlled group. This limitation has been pointed out by the authors. Second, as the authors pointed out, the study is not well designed, and patients may take some other interventions during the 6-year period, leading to long-term pain relief by BoNT-A biased. Authors should collect those factors, and perform the compounded analysis to remove all the bias. As authors pointed out, this is a convenience study (following a previous study), and not a designed study. Therefore, the statistical analysis is not valid (or at least not fully valid), and the conclusions from this study are questionable (especially on their long-term efficacy and reversibility of adverse effects.

Minor issues: English needs to be proofread (for example, in line 123, “than” should be “as”). Authors used a 95% confidence interval, however, in line 66 and line 75, authors claimed that no significant differences as p>0.005. Authors need to clarify that.

Author Response

Reviewer 2

Thank you for handling our manuscript. We revised it according to your comments and addressed each issue bellow, in a point-by-point manner. Our modifications are in track changes in the manuscript. The answers are in bold. We feel that by incorporating your suggestions in the revised version the clarity of our manuscript has improved. We would be glad to provide any further information needed.

First, the patient population is too small, and there is no placebo-controlled group. This limitation has been pointed out by the authors.

Reply: Thank you for observation. This limitation was discussed in the manuscript (lines 184-204; lines 231-237).

Second, as the authors pointed out, the study is not well designed, and patients may take some other interventions during the 6-year period, leading to long-term pain relief by BoNT-A biased. Authors should collect those factors and perform the compounded analysis to remove all the bias.

Reply: We understand your concern, but we do not consider that the study is not well-designed. Indeed, the study has limitations due to the long-term characteristic; however, it has relevant answers regarding the muscle conditions (adverse effects). In addition, we asked participants about other therapies used during this period, and they affirmed not using any other therapy specific for myofascial pain (only the behavioral therapies previously prescribed - lines 194-197). Therefore, it has to be pointed out that we must trust in the information delivered.

As authors pointed out, this is a convenience study (following a previous study), and not a designed study. Therefore, the statistical analysis is not valid (or at least not fully valid), and the conclusions from this study are questionable (especially on their long-term efficacy and reversibility of adverse effects.

Reply: We respect your opinion, but it has to be pointed out that this is not a ‘convenience study’; this study had a convenience sample (which is a different concept of ‘not designed’). It was always planned to follow these patients for as long as possible. But, as for all clinical studies, the major difficulty is to have adherence of patients for longer periods, especially if they did not see/feel differences. So, we agree that the patients that returned could be the ones with better outcomes, and for this reason we explained the caution on interpreting results regarding pain. However, data on muscle thickness is complete valid since it does not dependent on the effects of BoNT-A on pain.

Minor issues: English needs to be proofread (for example, in line 123, “than” should be “as”). Authors used a 95% confidence interval, however, in line 66 and line 75, authors claimed that no significant differences as p>0.005. Authors need to clarify that.

Reply: We apologize for the typo. P-value was corrected, and English was proofread.

Reviewer 3 Report

This is a well written manuscript. I do have several questions for the authors

1. While this study talks of myofascial pain it is really a study of TMD. The authors quote 45% of TMD have myofascial pain. That would imply 55% do not. Is the title misleading should it state TMD related pain.

2. The authors state that patient were injected 30-75 units in the masseters. how was this determined and was there a dose response to diminishment of pain.?? In addition did dose help determine duration of response and amount of muscle atrophy.

3. While the authors look at pain relief at 6 years due to Bont A, how do we know the pain relief was due to Bont A. i.e were patients taking other medications for pain control over the years.  Did they have surgical or other procedures, mouthguards etc. A comment regarding this would be beneficial.

4, Do the authors feels there is a sample bias in this study. Only 23% of patients were reevaluated at 6 years. Is there any follow up or information regarding discontinuation rates. Is it no possible the other 77% had no pain relief?? - a comment addressing this is needed.

Author Response

Reviewer 3

Thank you for handling our manuscript. We revised it according to your comments and addressed each issue bellow, in a point-by-point manner. Our modifications are in track changes in the manuscript. The answers are in bold. We feel that by incorporating your suggestions in the revised version the clarity of our manuscript has improved. We would be glad to provide any further information needed.

While this study talks of myofascial pain it is really a study of TMD. The authors quote 45% of TMD have myofascial pain. That would imply 55% do not. Is the title misleading should it state TMD related pain.

Reply: We appreciate the suggestion, but it should be mentioned that myofascial pain is one ‘type’ of temporomandibular disorder (TMD). TMD is an umbrella term that comprises masticatory muscle and joint conditions. As reported in the first sentence of our introduction, 45% of patients with TMD are diagnosed with myofascial pain, while the remaining 55% could have another diagnosis. In our study, as you can see in the Material and Methods section, we selected patients with the diagnosis of myofascial pain (alone or combined with other TMD diagnosis), which explain the Title. But, for attending your suggestion, we had added the term TMD in the title.

  1. The authors state that patients were injected 30-75 units in the masseters. how was this determined and was there a dose response to diminishment of pain.?? In addition, did dose help determine duration of response and amount of muscle atrophy.

Reply: Thanks for your questions: Doses were determined using the recommended range for each muscle by literature, since do not exist validated protocols. In our previous study, patients were divided in three groups of BoNT-A with different doses, and there was a significant reduction of perceived pain regardless of doses. Concerning muscle atrophy, our previous study also reported that there was a significant decrease in muscle thickness regardless of doses. All this information can be found in the previous study, which is published in TOXINS.

  1. While the authors look at pain relief at 6 years due to Bont A, how do we know the pain relief was due to Bont A. i.e were patients taking other medications for pain control over the years.  Did they have surgical or other procedures, mouthguards etc. A comment regarding this would be beneficial.

Reply: Thanks for your question. In fact, patients were asked about this issue, and as reported in the last paragraph of the manuscript, they referred that they used solely self-care strategies to control pain. We have addressed this important topic in lines 194-197.

4, Do the authors feels there is a sample bias in this study. Only 23% of patients were reevaluated at 6 years. Is there any follow up or information regarding discontinuation rates. Is it no possible the other 77% had no pain relief?? - a comment addressing this is needed.

Reply: Thanks for your proper suggestion and we totally agree with you. We had already addressed this issue as a limitation of our study in the last paragraph of the discussion section (lines 197-200).

Round 2

Reviewer 1 Report

Dear Authors, thank you for the corrections made. The manuscript is now in my opinion suitable for pubblication.

Best regards

Author Response

Thank you for handling our manuscript.   Best Regards

Reviewer 2 Report

While the authors explained the rationale in their response, this is a small trial, especially given that only 14 patients responded to their follow-up, and there is no placebo control arm. The conclusions should add that the results are preliminary, and cannot exclude there are placebo effects.

The Title should avoid using the abbreviation, rather “in myofascial temporomandibular disorders pain” rather than “myofascial pain TMD”.  

Author Response

Reviewer 2

Thank you for handling our manuscript. We revised it according to your comments and addressed each issue bellow, in a point-by-point manner. Our modifications are in track changes in the manuscript. The answers are in bold. We feel that by incorporating your suggestions in the revised version the clarity of our manuscript has improved. We would be glad to provide any further information needed.

- While the authors explained the rationale in their response, this is a small trial, especially given that only 14 patients responded to their follow-up, and there is no placebo control arm. The conclusions should add that the results are preliminary and cannot exclude there are placebo effects.

Reply: The suggestions of the reviewer were added in the conclusion section (lines 283-284).

- The Title should avoid using the abbreviation, rather “in myofascial temporomandibular disorders pain” rather than “myofascial pain TMD”.  

Reply: The abbreviation was changed accordingly (line 3).

Reviewer 3 Report

thank you for your responses to all previous queries. 

Author Response

(The authors gave the same response as above.)
